# Differential Diagnosis of Major Depressive Disorder and Bipolar Disorder: Genetic and Hormonal Assessment and the Influence of Early-Life Stress

**DOI:** 10.3390/brainsci12111476

**Published:** 2022-10-31

**Authors:** Itiana Castro Menezes, Cristiane von Werne Baes, Fernanda Viana Fígaro-Drumond, Brisa Burgos Dias Macedo, Ana Carolina Bueno, Riccardo Lacchini, Marcelo Feijó de Mello, Margaret de Castro, Mario Francisco Juruena

**Affiliations:** 1Department of Neurosciences and Behavior, Ribeirao Preto Medical School, University of Sao Paulo, Ribeirao Preto 14015-130, Brazil; 2Department of Psychiatric Nursing and Human Sciences, Ribeirao Preto College of Nursing, University of Sao Paulo, Ribeirao Preto 14040-902, Brazil; 3Department of Pediatrics, Ribeirao Preto Medical School, University of Sao Paulo, Ribeirao Preto 14015-130, Brazil; 4Department of Psychiatry, Federal University of Sao Paulo, Sao Paulo 04023-062, Brazil; 5Department of Internal Medicine, Ribeirao Preto Medical School, University of Sao Paulo, Ribeirao Preto 14015-130, Brazil; 6Department of Psychological Medicine, Institute of Psychiatry, Psychology and Neuroscience, King’s College London, London SE5 8AF, UK

**Keywords:** depression, bipolar, biomarker, mineralocorticoid receptor (MR), glucocorticoid receptor (GR), polymorphism (SNP), cortisol, aldosterone, early-life stress (ELS)

## Abstract

Few studies have assessed biomarkers for the differentiation of major depressive disorder (MDD) and bipolar disorder (BD). However, some elements of depression such as hormones and receptors of the renin–angiotensin–adrenal system (RAAS), the hypothalamus–pituitary–adrenal (HPA) axis, and history of early-life stress (ELS) could be considered for differential diagnosis. Therefore, this study aimed to assess aldosterone and cortisol levels, MR and GR gene polymorphisms, and ELS as potential biomarkers for differentiating MDD and BD. This study presents a case–control design. Groups comprised samples for genetic, cortisol, and aldosterone analysis: healthy control (HC; *n* = 113/97/103), MDD (*n* = 78/69/67) and BD (*n* = 82/68/65) subjects. Furthermore, all subjects were assessed for diagnostic screening, the severity of depression, and history of ELS by applying MINI-PLUS, GRID-HDRS, and CTQ, respectively. In addition, genotype and allelic frequencies of GR (N363S, R22/23K and BclI) and MR (MI180V and -2G/C) polymorphisms were evaluated via PCR. Our findings demonstrate that basal aldosterone levels may be a biomarker for differentiating BD and MDD. Furthermore, ELS affects the HPA axis in BD, cortisol may be considered a biomarker for distinguishing BD and MDD, but only in the absence of ELS, and, finally, history of ELS and MR-2G/C variant alleles are factors that contribute to the severity of depressive symptoms in MDD and BD.

## 1. Introduction

Depression is a common, recurrent, and chronic disorder that affects subjects systemically, impairing their mood, as well as autonomic and cognitive functions [1]. By 2017, according to the World Health Organization (WHO), depression has become the most incapacitating disorder worldwide [2], impairing personal, social, and labor well-being.

Major depressive disorder (MDD) and Bipolar Disorders (BD) tend to have similar symptoms. However, about 12–20% of BD patients are misdiagnosed as having MDD in the first year of treatment. In the National Comorbidity Survey Replication (NCS-R) sample, 16.2% of MDDs were lifelong and 6.6% were experienced in the last 12 months before the interview. An estimated 4.4% of U.S. adults experience BD at some time in their lives [1,2,3,4,5]. Additionally, 40% to 50% of bipolar patients—type II or bipolar spectrum—take about ten years to obtain the correct diagnosis and treatment [4]. During these years of mistreatment, bipolar depression usually becomes chronic and more severe [5].

Affective disorders are multifactorial, as biological and environmental factors contribute to their etiology. The etiologic factors of unipolar and bipolar depression can be both genetic and environmental. Among the genetic factors, we can mention alterations in the functioning of the hypothalamic–pituitary–adrenal (HPA) axis, as well as in the renin–angiotensin–aldosterone system (RAAS). The genes of the receptors of these axes and their hormones (cortisol and aldosterone) are the study targets of the present research [6,7]. Both HPA axis and RAAS are mediated by the limbic system and there is an interaction between them. [7,8]. Aldosterone and cortisol bind to mineralocorticoid receptors (MR) and glucocorticoid receptors (GR) to exert their actions on the RAAS and HPA axis, respectively. Inadequate levels of aldosterone and cortisol and MR/GR imbalance may impair subjects’ coping, raising their vulnerable phenotype to disorders related to stress exposition [8,9]. Thus, MR and GR gene polymorphisms could have a substantial role as biological factors related to depressive diseases in vulnerable subjects. Indeed, MR and GR gene polymorphisms have been associated with altered HPA axes and imbalanced RAAS functioning [10,11]. Among the environmental factors associated with depression, exposure to early-life stress (ELS) is a key factor [6]. The present study assessed MR and GR genes polymorphisms in patients with MDD and BD and healthy controls (HC).

ELS has been defined as a highly stressful and/or traumatic situation in which a person was exposed during childhood and/or adolescence. It may be highlighted as an extrinsic etiological factor for the development of mood disorders. Several studies observed neuroendocrine and epigenetic changes in patients exposed to ELS, which contributed to the development of psychiatric disorders. In humans, during childhood and adolescence, the central nervous system presents its most plastic period [6,12,13], and subjects exposed to ELS may acquire a more vulnerable psychiatric profile in adulthood.

Therefore, this study aimed to assess aldosterone and cortisol levels as well as MR and GR gene polymorphisms as possible intrinsic factors, and ELS as a putative extrinsic factor of depressive disorder development. In addition, we also aimed to assess these parameters as potential biomarkers for differentiating MDD and BD.

## 2. Materials and Methods

### 2.1. Participants

This study follows a case–control design. The genetic assessment sample comprised 273 subjects (*n* = 113 healthy controls (HC); *n* = 78 MDD patients; and *n* = 82 bipolar patients). The sample size was estimated by applying the software Power for Genetic Association Analysis (PGA) (available online on https://dceg.cancer.gov/tools/design/pga. Accessed in 20 July 2014), taking into account some variables such as the prevalence of affective disorders in the Sao Paulo Metropolitan Area (11%) [14], an alfa error of 0.05, and 80% power for detecting significant differences. The sample size for cortisol blood analysis comprised 234 subjects (*n* = 97 HC, *n* = 69 MDD, *n* = 68BD) and 235 subjects were considered for aldosterone blood analysis (*n* = 103 HC, *n* = 67 MDD, *n* = 65 BD). Participants of both genders ranging from 18 to 65 years old were recruited for the study.

Two different psychiatrists clinically evaluated the participants. MINI-PLUS [15] was applied for confirming or excluding MDD or BD diagnosis and other psychiatric disorders, according to DSM-5 [1]. Additionally, the 21-item GRID-Hamilton Depression Rating Scale (GRID-HAM-D21) was used to assess the severity of depressive symptoms [16]. None of the BD patients were in a manic or hypomanic state. All patients, MDD and BD, had been in stable drug therapy for the last 8 weeks. We also evaluated all subjects for ELS by applying the Childhood Trauma Questionnaire (CTQ) [17]. As a criterion for defining the presence of ELS, we considered the classification of at least one subtype of abuse or neglect as moderate to severe or severe to extreme in the CTQ scores according to the definition by Bernstein [18]. Participants who obtained scores equivalent to the classification of absence of ELS to minimal, or low to moderate for the 5 subtypes of ELS assessed by the CTQ were classified as subjects without ELS.

The exclusion criteria for the HC group were: having psychiatric/neurological disorders, taking any psychotropic medication or having severe clinical disease during life, or having first-degree relatives with psychiatric/neurological disorders. Exclusion criteria for the patients were: history of hypersensitivity to corticosteroids or steroid use, heavy smoking, viral illness during the preceding two weeks, pregnancy or lactation, alcohol or drug abuse/dependence, and significant physical illness. We also excluded patients with intellectual disabilities, psychotic symptoms unrelated to their depressive disorder, or an organic cause for depression.

The present study was approved by the Ethics Committee at the Research of Clinical Hospital of Ribeirao Preto Medical School of University of Sao Paulo (HCFMRP/USP; CAAE 31124114.3.0000.5440) and followed the guidelines and standards for research involving human subjects of the Brazilian National Health Council (CNS) resolution 466/12-and Declaration of Helsinki [19].

### 2.2. Hormonal Assessment

Peripheral blood samples were collected around 9 am. to assess basal cortisol and aldosterone levels. Samples were centrifuged at 3000 rpm at 4 °C and stored at −80 °C until the day of the assay. For cortisol analysis, a duplicate of the 50 µL serum sample was assessed by radioimmunoassay (RIA), as previously published [20]. All the results were provided in microgram/deciliter (mg/dL) units and later converted to nanomole/liter (nmol/L), according to the international system of units (SI). Intra and interassay variabilities were 5.9% and 14.3%, respectively. Aldosterone analysis was assessed in duplicates, using 100 µL of blood serum, via the chemiluminescent immunoassay (CLIA) method using the LIAISON^®^ kit (DiaSorin; Saluggia, Italy). All the results are provided in nanogram/deciliter (ng/dL) units and later converted according to SI to picomole/liter (pmol/L). Cortisol and aldosterone serum samples were assessed at the Laboratory of Endocrinology-HCFMRP/USP. 

### 2.3. Genetic Assessment

Peripheral blood samples from all participants were collected using sterile EDTA blood tubes (Vacutainer^®^, BD; Curitiba, Brazil) and stored at −20 °C. DNA was extracted from total peripheral blood samples using the QIAamp^®^ DNA Mini Kit (QIAGEN; Hilden, Germany). TaqMan allelic discrimination assays were performed using real-time PCR (qPCR; Applied Biosystems, ThermoFisher Scientific; Waltham, MA, USA) for GR (N363S or rs56149945; R22/23K or rs6190; and BclI or rs41423247), and MR (MI180V or rs5522, and -2 G/C or rs2070951) polymorphisms were genotyped. Each DNA sample was assessed in duplicate. Each one of the 96-well plates was fulfilled with a total volume of 6.74 µL, of which 3.125 µL was GoTaq^®^ Probe qPCR Master Mix (Promega; Madison, WI, USA), 0.156 µL was primer–probe (designed by Applied Biosystems, ThermoFisher Scientific; Waltham, MA, USA), and 3.5 µL was DNA sample (20 ng/µL). Each reaction plate contained 2 wells used as a positive control (3.5 µL DNA) and 2 wells used as a negative control (3.5 µL free-DNAse/RNAse purified water). The amplification reaction and allelic discrimination via Real-Time System 7500 apparatus (Applied Biosystems, ThermoFisher Scientific; Waltham, MA, USA) followed the steps: pre-reading phase for 1 min; 95 °C for 10 min (step 1); 92 °C for 15 sec (step 2); and 60 °C for 1 min (step 3) with data acquisition. Steps 2 and 3 were repeated for 40 cycles. Post-read qPCR was the last step (1 min). Applied Biosystems 7500 and 7500 FAST software, v. 2.0.4 (ThermoFisher Scientific; Waltham, MA, USA), were applied to assess allelic discrimination during a qPCR reaction.

### 2.4. Data Analysis

Statistical analysis was performed using the Statistical Package for Social Science software version 20.0 (SPSS, v.20.0). Three-way analysis of covariance (ANCOVA) was applied for evaluating basal hormonal levels, considering three isolated independent factors (groups (G); polymorphisms (SNPs); ELS (ELS)) or their interactions (GxELS; GxSNP; ELSxSNP; GxELSxSNP), and adjusting mean and standard error (s.e.) for the covariables (age, depressive symptoms intensity score, oral contraceptive, lithium, anticonvulsant, antipsychotic). The SIDAK post hoc test was applied to detect differences when comparing three factors. Hormonal levels and scores of GRID-HAM-D21 were converted to a logarithmic scale (log10) during data analysis to obtain a normal distribution. For qualitative data and deviation from the Hardy–Weinberg Equilibrium (HWE), an exact or asymptotic chi-square (X^2^) test was applied. Data normality was assessed by applying the Levene test. The significance level established was 0.05.

## 3. Results

### 3.1. Sociodemographic Data

Table 1 shows sociodemographic characteristics, history of ELS, and genotype frequency of MR and GR polymorphisms in HC, MDD and BD groups. In Table 1, characteristics are described by “genetics sample”, “cortisol sample” and “aldosterone sample”. GRID-HAM-D21, CTQ, and ELS (%) parameters were lower in HC compared with the MDD and BD groups. We found no difference when these three groups were compared for genotype frequencies of MR MI180V, MR-2 G/C, GR N363S, GR R22/23 K, and GR BclI. For more information about the medication taken by the participants, see Appendix A (Table A1). 

### 3.2. Hormonal Levels

Cortisol levels showed a significant difference in these situations: (a) under the effect of G x ELS interaction, in which the BD sample presented lower cortisol levels than the MDD and HC groups—Figure 1a and Appendix B (Table A2); (b) in the intragroup analysis, in which BD subjects with ELS presented higher levels of cortisol than BD without ELS (Figure 1a and Appendix B—(Table A2)); and (c) when compared with BclI GC carriers, the BclI CC genotype provided higher cortisol levels for all carriers (Figure 1c and Table 2). BD patients presented lower aldosterone levels than the MDD group considering ELS (Figure 1b and Appendix B), and MR and GR polymorphisms (Table 3).

### 3.3. The Severity of Depressive Symptoms

Depressive symptoms may worsen in the presence of ELS for MDD or BD patients - Table 4 and Appendix C (Figure A1a). Regarding SNPs, we observed that the presence of MR-2 G/C variant alleles might increase the intensity of depressive symptoms of unipolar patients, mainly in homozygosis—Table 4 and Appendix C (Figure A1b). Additionally, the MR I180V G variant allele showed increasing depressive symptoms for unipolar and bipolar groups—Table 4 and Appendix C (Figure A1b).

## 4. Discussion

This study evaluated cortisol and aldosterone levels, GR and MR polymorphisms, and the history of ELS as a modulating factor of MDD and BD. Our main findings pointed out: (a) lower aldosterone levels in BD compared with MDD patients; (b) in the absence of ELS, cortisol levels were reduced in BD compared with the HC and MDD groups; (c) a significant difference in cortisol levels was observed when comparing BD subjects with ELS to BD subjects without ELS; (d) increased cortisol levels were found in subjects with BclI CC genotype; (e) reduced aldosterone levels were found in the presence of MR MI180V G allele; (f) more severe depressive symptoms were found in unipolar and bipolar patients carrying MR-2 G/C variant alleles or ELS. Thus, our data reinforce that the HPA axis and RAAS imbalanced functioning and ELS may be involved in depression, as previously addressed [10,12,13,18,21]. In addition, our data also contributed to clarifying the involvement of GR and MR polymorphisms in MDD and BD, mainly for differential diagnosis, which is a key finding since articles focusing on this issue are still scarce [22,23].

Most studies investigating biomarkers for the diagnosis or prognosis of depression have a heterogeneous sample composed of unipolar and bipolar patients; bipolar patients in different states, such as depressed, mixed, manic, and euthymic; and MDD or BD patients, without controlling the intensity of symptoms. Nevertheless, according to the literature, gene and protein expression, methylation and, thus, hormonal levels may change depending on the mood state in BD and the severity of depression in MDD or BD [24,25,26,27]. Therefore, one strength of our study is that the sample was composed of a similar proportion of unipolar and bipolar patients, concerning the severity of depressive symptoms. Additionally, our data analysis was controlled for medicine intake, age, and intensity of depressive symptoms.

In our study, the reduction in morning basal cortisol levels in BD compared with the HC and MDD groups was detected only in the absence of ELS. Moreover, by intragroup analysis, it was possible to observe increased cortisol levels in BD patients with ELS compared with those without ELS (Figure 1a). Yehuda et al. [27] suggest that BD and PTSD patients present an enhanced GR number. In contrast, MDD presents a reduction in GR number. Some studies have shown that morning basal cortisol [28,29] and 24 h urinary cortisol [27] levels may be raised in depressed patients compared with controls. Our data corroborated Feng et al.’s [30] findings, showing that BD patients had lower cortisol levels than MDD patients. Many studies in the literature reinforce other abnormalities in cortisol regulation. Havermans et al. [31], after a multilevel regression analysis, concluded that despite the lack of difference in basal cortisol levels between control and bipolar patients, the latter showed flattened diurnal slopes and more significant cortisol fluctuations over successive measures. Additionally, patients with a higher incidence of previous depressive episodes presented higher overall cortisol levels, reduced cortisol reactivity to daily adverse events, and flattened diurnal slopes compared with those with a lower incidence of depressive disease. Huang et al. [32] found a lower cortisol awakening response (CAR) in BD patients in a depressed state compared with those in a manic state. Altogether, these studies, including ours, have considered cortisol as a factor for distinguishing depression severity in BD, suggesting that cortisol levels may be used as a potential biomarker for distinguishing states in BD courses.

Traumatic stress in the early stages of development alters physiological, biochemical and genetic pathways. Regarding studies on cortisol and stress, Mazer et al. [33] reported a negative correlation between cortisol levels and exposition to ELS in BD patients who suffered sexual abuse. The higher the exposition scores, the lower the cortisol levels. Indeed, a history of ELS is associated with more severe depressive states [12,21]. The earlier the abuse and neglect, the more severe and unstable the course of BD [34].

In the present study, we evaluated the genotype frequencies of the most common SNPs related to GR and MR receptors: MR MI180V, MR-2 G/C, GR N363S, GR R22/23 K, and GR BclI. No differences were observed in genotype frequencies among HC, MDD and BD subjects. However, we observed higher cortisol levels in subjects carrying the BclI CC genotype compared with BclI GC, regardless of the group (Figure 1b). Indeed, Kumsta et al. [35] and Wüst et al. [36] found that the BclI GG genotype was associated with a diminished cortisol response in males, but this was increased in women after the psychosocial stress test. On the other hand, Velders et al. [37] observed the interaction between prenatal psychological symptoms and BclI (rs41423247), resulting in decreased child cortisol reactivity. Based on these data, we may infer that BclI genotypes influence cortisol levels differently, mainly depending on gender, which could explain the divergent findings.

In our study, considering only subjects without ELS, cortisol levels in bipolar patients were lower than in MDD and HC groups. Moreover, GR SNP R22/23 K T presented an effect on cortisol levels compared with GR R22/23 K C (Table 4). Unfortunately, our sample size for this SNP was not large enough to infer its real effect. However, it is essential to point out that GR R22/23 K T has been previously associated with depressive disease, even considering its rarity as a variant allele in the general population. Bet et al. [38] evaluated the effect of the R22/23 K TT genotype in a small sample of depressed subjects with a history of ELS. They found that this genotype increased the chance of developing depression by 12 times compared with non-carriers. GR R22/23 K TT is also associated with recurrent depression [39,40]. Further studies are necessary to clarify the role of the GR R22/23 K TT genotype in depressive disorders with or without ELS.

Another putative factor for the differential diagnosis of BD and MDD is aldosterone serum levels. Emanuele et al. [41] observed aldosterone serum levels 2.77 times higher in MDD patients compared with controls. Although we did not find a difference in aldosterone levels in bipolar or unipolar patients compared with the controls, our data showed that bipolar patients had lower aldosterone levels than MDD patients (Figure 1b), regardless of the presence or absence of ELS. Hullin et al. [42], on the other hand, found higher serum aldosterone levels in BD patients in a manic state compared with those in the early stage of depression. The effect of ELS on aldosterone levels was related to the interaction of the ELS and GR N363S phenotype (Table 3).

Aldosterone levels have not only been linked to depression diagnosis, but also identified as a biomarker for depression prognosis. Murck et al. [43] assessed basal salivary aldosterone levels in MDD subjects and found that patients with remission presented a reduction in aldosterone levels after 6 weeks of treatment. Büttner et al. [44] also assessed salivary aldosterone in MDD subjects six months after depression. They found higher levels of aldosterone resulting in sodium loss, higher rates of aldosterone/cortisol, and lower blood pressure, suggesting that aldosterone levels could be a predictive factor for a worse prognosis in MDD [44].

Holsboer et al. [45] studied unipolar and bipolar patients after the dexamethasone suppression test (DST) and observed enhanced aldosterone levels in HC compared with depressive subjects. The aldosterone enhancement after the DST test would be a consequence of the suppression of other mineralocorticoids such as l-desoxycorticosterone and corticosterone (both ACTH-dependent hormones). These authors concluded that aldosterone levels have an essential role in depression. Häfner et al. [46] have also shown higher levels of basal serum aldosterone and systemic arterial hypertension in unipolar and bipolar depressed patients, suggesting aldosterone and its MR can be associated with depression. However, further studies are needed to determine whether they are involved in the etiology of unipolar and bipolar depression.

In our study, the MR-2 G/C polymorphism was associated with a negative effect on the intensity of depressive symptoms, while MR MI180V showed only a small tendency (Table 4). These variant alleles may worsen depressive symptoms, as Kuningas et al. [47] observed in elders carrying the MR MI180V allele variant. Our findings on MR-2 G/C G alleles and their links to depression are in accordance with most literature studies relating impaired MR expression and depression [48]. Our data corroborated to Taylor et al.’s [49] study, which showed that CC carriers have a half-chance of developing depression and suicide ideation compared with GG carriers.

ELS, in our sample, was more prevalent in MDD and BD groups than in the HC group, but there was no difference in ELS prevalence between MD and BD groups. This fact corroborates with the current literature showing the harmful effect of ELS in MDD and BD patients, and the association of ELS with the presence of more severe depression symptoms [6,12,21,49,50]. In addition, subjects with ELS also have higher chances of developing other psychiatric disorders, such as substance abuse, anxiety, and disruptive, dissociative, eating, psychotic, and personality disorders [6].

### Limitations and Strengths

This study presented some limitations. Depressed patients had already been treated with pharmacological drugs, and most of them presented severe cases of depression, considering that our service as a tertiary one. Genetic, aldosterone and cortisol evaluation groups did not have the same sample size. We prioritized the biological sample for the genetic study, and a greater sample size is required to estimate the differences among the HC, MDD and BD groups with a higher power. The sample sizes for the MR MI180V, MR-2 G/C, and GR BclI polymorphisms can be considered adequate; to evaluate GR N363S and GR R22/23 K polymorphisms, it would be necessary a greater sample size for a better estimation of the power of the dependent variables. Thus, the study has a moderate sample size with a detailed psychiatric screening of all subjects enrolled to ensure a reduction in selection bias. To transpose our findings for medical application, it would be essential to replicate and validate the results in other populations.

## 5. Conclusions

Our main findings suggest that aldosterone serum levels may be applied as a biomarker for differentiating BD and MDD. In contrast, cortisol serum levels may be applied only in the absence of ELS. Furthermore, besides the presence of ELS, which significantly affects the HPA axis in bipolar patients, MR-2 G/C variant alleles are also a factor for worsening depressive symptoms in MDD and BD. Thus, altered genetic and hormonal compounds of the HPA axis and the RAAS may be involved in MDD and BD, and may be used as potential biomarkers for the differential diagnosis of depression severity. Moreover, a new possibility of investigation could be assessing aldosterone as a potential biomarker of treatment response in a clinical trial. Additionally, ELS must be considered an aggravating factor of depression severity in BD and MDD.

## Figures and Tables

**Figure 1 brainsci-12-01476-f001:**
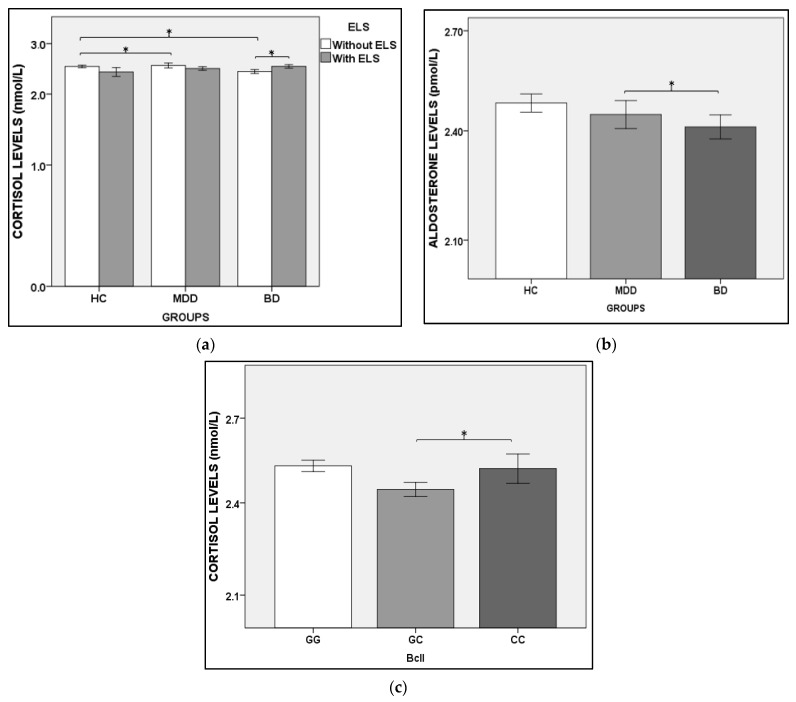
Hormonal levels presented in logarithmic scale (*Y*-axis). (**a**) Mean cortisol levels in HC, MDD and BD groups, in the presence (white bars) and absence (gray bars) of ELS; (**b**) mean aldosterone levels in HC, MDD, and BD groups; (**c**) Mean cortisol levels compared with BclI genotype. Note: BD: bipolar disorder; ELS: early-life stress; HC: health control; MDD: major depression disorder; * *p* < 0.05; error bars: +/−1 standard error.

**Table 1 brainsci-12-01476-t001:** Sociodemographic characteristics, parameters related to depression symptoms, history of early-life stress, and genotype frequency of mineralocorticoid and glucocorticoid receptor polymorphisms in healthy controls, as well as unipolar and bipolar subjects.

	Genetic Sample			Statistics	Cortisol Sample			Statistics	Aldosterone Sample			Statistics
	HC	MDD	BD		HC	MDD	BD		HC	MDD	BD	
**AGE** (mean ± sd)	37.20 (15.27)	41.17 (12.34)	40.18 (12.59)	F = 2.098; *p* = 0.619	35.98 (14.46)	40.90 (12.62)	39.73 (12.67)	**F = 3.083;** ***p* = 0.048 a,b**	35.47 (14.19)	40.61 (12.33)	40.0 (12.36)	**F = 3.939;** ***p* = 0.021 a,b**
**SEX** (F/M) (%)	69.91/ 30.11	79.49/ 20.51	80.49/19.51	X^2^ = 3.699; *p* = 0.157	70.1	79.7	80.6	X^2^ = 3.264; *p* = 0.207	70.9	80.6	80.0	X^2^ = 2.840; *p* = 0.258
**BMI** (Kg/m^2^)	26.52 (5.80)	28.33 (6.42)	27.92 (6.33)	F = 1.985; *p* = 0.142	26.37	27.90	27.98	F = 1.521; *p* = 0.221	26.27 (5.83)	27.55 (6.16)	27.77 (6.50)	F = 1.307; *p* = 0.273
**GRID-HAM-D21**	0.5 (0.907)	17.78 (9.76)	19.03 (9.88)	**X^2^ = 257.839;** ***p* < 0.001 a,b**	0.51 (0.96)	17.84 (9.92)	19.05 (9.61)	**X^2^ = 223.576;** ***p* < 0.001 a,b**	0.52 (0.94)	18.46 (9.90)	19.85 (9.36)	**F = 87.694;** ***p* < 0.001 a,b**
**CTQ**	30.55 (8.64)	45.32 (16.60)	50.68 (24.12)	**X^2^ = 31.147;** ***p* < 0.001 a,b**	30.31 (8.73)	46.07 (16.85)	50.58 (24.40)	**X^2^ = 31.147;** ***p* < 0.001 a,b**	30.03 (8.40)	46.01 (17.37)	49.63 (24.08)	**F = 32.89;** ***p* < 0.001 a.b**
**ELS (%)**	7.24	56.84	59.49	**X^2^ = 71.394;** ***p* < 0.001 a,b**	4.3	58.8	59.1	**X^2^ = 71.400;** ***p* < 0.001 a,b**	5.0	56.7	59.4	**X^2^ = 70.103;** ***p* < 0.001 a,b**
**MR MI180V AA/AG/GG**	79.65/ 18.58/ 1.77	79.79/ 19.23/ 1.28	85.37/ 14.63/ 0.0	X^2^ = 2.215; *p* = 0.757	78.4/ 19.6/ 2.1	79.7/ 18.8/ 1.4	85.3/ 14.7/ 0.0	X^2^ = 2.173; *p* = 0.765	78.6/ 19.4/ 1.9	77.6/ 20.9/ 1.5	86.2/ 13.8/ 0.0	X^2^ = 2.605; *p* = 0.693
**MR-2 G/C CC/CG/GG**	28.32/ 45.13/ 26.55	21.79/ 48.72/ 29.49	20.73/ 50.0/ 29.27	X^2^ = 1.840; *p* = 0.768	26.8/ 47.4/ 25.8	24.6/ 46.4/ 29.0	22.1/ 51.5/ 26.5	X^2^ = 0.736; *p* = 0.948	28.2/ 45.6/ 26.2	20.9/ 49.3/ 29.9	21.5/ 50.8/ 27.7	X^2^ = 1.595; *p* = 0.815
**GR N363S TT/TC/CC**	95.59/ 4.04/ 0.0	93.59/ 5.41/ 0.0	93.92/ 5.08/ 0.0	X^2^ = 6.251; *p* = 0.131	98.0/ 1.0/ 1.0	92.8/ 7.2/ 0.0	92.6/ 7.4/ 0.0	X^2^ = 6.324; *p* = 0.124	99.0/ 1.0/ 0.0	92.5/ 7.5 /0.0	93.8/ 6.2/ 0.0	X^2^ = 4.932; *p* = 0.103
**GR R22/23K** **CC/CT/TT**	97.32/ 2.68/ 0.0	98.72/ 1.28/ 0.0	98.78/ 1.22/ 0.0	X^2^ = 0.743; *p* = 0.737	99.0/ 1.0/ 0.0	98.6/ 1.4/ 0.0	98.5/ 1.5/ 0.0	X^2^ = 0.083; *p* = 1.0	98.0/ 2.0/ 0.0	98.5/ 1.5/ 0.0	98.5/ 1.5/ 0.0	X^2^ = 0.068; *p* = 1.0
**GR BclI GG/GC/CC**	45.79/ 41.59/ 10.62	37.18/ 55.13/ 7.69	51.22/ 37.81/ 10.97	X^2^ = 5.460; *p* = 0.245	45.4/ 43.3/ 9.1	31.8/ 30.7/ 6.5	52.9/ 38.2/ 8.8	X^2^ = 3.455; *p* = 0.492	46.6/ 41.7/ 11.7	37.3/ 56.7/ 6.0	53.8/ 38.5/ 7.7	X^2^ = 6.496; *p* = 0.166

BD: bipolar disorder; BMI: body mass index; CTQ: Childhood Trauma Questionnaire; ELS: early-life stress; GR: glucocorticoid receptor gene; GRID-HAM-D21: GRID–Hamilton Depression Rating Scale; HC: healthy controls; MDD: major depressive disorder; MR: mineralocorticoid receptor gene; sd: standard deviation; post hoc test (SIDAK test): * *p* < 0.05. a: significant difference between MDD patients and healthy controls; b: significant difference between BD patients and HC subjects.

**Table 2 brainsci-12-01476-t002:** Cortisol levels (nmol/L) in healthy controls, MDD and BD patients concerning history of early-life stress and mineralocorticoid and glucocorticoid receptor polymorphisms.

	HC		UNIPOLAR		BIPOLAR		
	Without ELS	With ELS	Without ELS	With ELS	Without ELS	With ELS	
	(Mean ± Standard Error)						STATISTICS
**MR MI180V AA**	348.34 (1.10)	244.34 (1.46)	318.42 (1.13)	311.17 (1.13)	207.97 (1.15)	322.11 (1.13)	**G**: F = 2.295; *p* = 0.104; **ELS**: F = 0.389; *p* = 0.534; **SNP**: F = 0.975; *p* = 0.379; **GxELS**: F = 1.867; *p* = 0.158 *BD (AA: without ELS < with ELS) *;* **GxSNP**: F = 0.620; *p* = 0.603; **ELSxSNP**: F = 0.256; *p* = 0.614 *Without ELS (AA: BD < HC) *;* **GxSNPxELS**: F = 0.659; *p* = 0.519.
**MR MI180V AG**	322.66 (1.20)	465.59 (1.71)	430.53 (1.32)	337.29 (1.28)	223.87 (1.36)	342.77 (1.32)	
**MR MI180V GG**	--	244.91 (1.71)	--	663.74 (1.71)	--	--	
**MR-2 G/C CC**	322.85 (1.14)	138.68 (1.71)	339.62 (1.20)	333.43 (1.27)	221.31 (1.37)	322.85 (1.26)	**G**: F = 1.674; *p* = 0.191; **ELS**: F = 0.000; *p* = 0.995 *BD (without ELS < with ELS) *;* **SNP**: F = 2.181; *p* = 0.117; **GxELS**: F = 2.846; *p* = 0.061 *(Without ELS: BD < MDD) *;* **GxSNP**: F = 0.726; *p* = 0.576; **ELSxSNP**: F = 1.102; *p* = 0.335; **GxSNPxELS**: F = 0.427; *p* = 0.789 *BD (CG: without ELS < with ELS).*
**MR-2 G/C CG**	379.31 (1.13)	437.52 (1.46)	343.56 (1.23)	373.25 (1.16)	215.28 (1.22)	347.57 (1.16)	
**MR-2 G/C GG**	309.03 (1.17)	235.50 (1.71)	333.43 (1.25)	254.10 (1.20)	204.17 (1.21)	2.45 (1.26)	
**GR N363S TT**	343.56 (1.10)	286.42 (1.31)	322.11 (1.13)	326.89 (1.12)	213.30 (1.14)	324.34 (1.12)	**G**: F = 1.261; *p* = 0.286; **ELS**: F = 0.076; *p* = 0.783; **SNP**: F = 0.593; *p* = 0.554; **GxELS**: F = 2.743; *p* = 0.068; **GxSNP**: F = 0.264; *p* = 0.769; **SNPXELS**: F = 0.739; *p* = 0.391; **GxSNPXELS**: F = 0.935; *p* = 0.335 *Without ELS (TT: BD < HC) ** *BD (TT: without ELS < with ELS) **.
**GR N363S TC**	588.84 (1.71)	--	506.99 (1.45)	237.14 (1.37)	180.30 (1.73)	286.42 (1.48)	
**GR N363S CC**	238.23 (1.70)	--	--	--	--	--	
**GR R22/23 K CC**	349.14 (1.10)	287.08 (1.31)	344.35 (1.12)	319.15 (1.12)	195.88 (1.14)	324.34 (1.12)	**G**: F = 0.821; *p* = 0.442; **ELS**: F = 0.424; 0.516; **GxELS**: F = 4.347; *p* = 0.015 * *(BD < HC*; BD < MDD *)* **SNP**: F = 0.403; *p* = 0.526; **GxSNP**: F = 3.179; *p* = 0.044 * *BD (CC: without ELS < with ELS) ** *BD (without ELS: CC < CT) ** *Without ELS (CC: BD < MDD)* *Without ELS (BD < MDD) *.*
**GR R22/23 K CT**	234.42 (1.69)	--	250.61 (1.69)	--	722.770 (1.70)	--	
**GR R22/23 K TT**	--	--	--	--	--	--	
**GR BclI GG**	406.44 (1.12)	422.67 (1.69)	350.75 (1.19)	328.85 (1.67)	244.34 (1.18)	311.89 (1.16)	**G**: F = 0.927; *p* = 0.398; **ELS**: F = 0.038; *p* = 0.845 *without ELS (HC > BD) ** *without ELS (MDD > BD) ** *BD (without ELS< with ELS) *;* **SNP**: F = 3.346; *p* = 0.038 * *(CC > GC) ** *BD (CC > GC) *;* **GxELS**: F = 1.530; *p* = 0.220; **GxSNP**: F = 0.432; *p* = 0.785 **ELSxSNP**: F = 0.051; *p* = 0.950; **GxSNPxELS**: F = 0.151; *p* = 0.860
**GR BclI GC**	301.30 (1.12)	257.63 (1.36)	307.61 (1.18)	305.50 (1.16)	175.79 (1.20)	271.02 (1.208)	
**GR BclI CC**	380.19 (1.25)	--	442.59 (1.445)	358.92 (1.48)	--	477.53 (1.28)	

Means and standard error (s.e.) were adjusted for covariables (age, depressive symptom intensity score, oral contraceptive, lithium, anticonvulsant, antipsychotic). HC: healthy controls; ELS: early-life stress as independent factor; G: groups as independent factor; GR: glucocorticoid receptor gene; MR: mineralocorticoid receptor gene; SNP: polymorphism independent factor; GxELS: interaction between groups and early-life stress; GxSNP: interaction between groups and polymorphisms; GxSNPxELS: interaction among groups, polymorphisms, and early-life stress. Post hoc test (SIDAK): * *p* < 0.05.

**Table 3 brainsci-12-01476-t003:** Aldosterone levels (pmol/L) in HC, MDD and BD patients concerning ELS, and mineralocorticoid and glucocorticoid receptor polymorphisms.

	HC		UNIPOLAR		BIPOLAR		
	Without ELS	With ELS	Without ELS	With ELS	Without ELS	With ELS	
	(Mean ± Standard Error)						STATISTICS
**MR MI180V AA**	329.61 (1.14)	402.72 (1.50)	317.69 (1.13)	286.42 (1.19)	273.53 (1.20)	203.24 (1.18)	**G**: F = 3.104; *p* = 0.048 * *(BD < MDD) ** **ELS**: F = 0.342; *p* = 0.560; **SNP**: F = 1.138; *p* = 0.323; **GxELS**: F = 0.151; *p* = 0.860; **GxSNP**: F = 0.596; *p* = 0.619 **SNPxELS**: F = 0.063; *p* = 0.803; **GxSNPxELS**: F = 1.144; *p* = 0.321 *Without ELS (AG:BD<HC) *.*
**MR MI180V AG**	391.74 (1.26)	181.97 (2.01)	299.22 (1.38)	300.61 (1.37)	123.59 (1.50)	162.93 (1.50)	
**MR MI180V GG**	--	222.33 (2.00)	--	291.74 (1.99)	--	--	
**MR-2 G/C CC**	279.25 (1.18)	313.32 (1.97)	295.12 (1.27)	375.84 (1.48)	216.77 (1.49)	272.27 (1.34)	**G**: F = 2.102; *p* = 0.126; **ELS**: F = 0.575; *p* = 0.450; **SNP**: F = 0.867; *p* = 0.422; **GxELS**: F = 0.099; *p* = 0.906; **GxSNP**: F = 1.297; *p* = 0.274 *CG (BD < MDD)*/MDD (GG < CG) *;* **SNPXELS**: F = 1.064; *p* = 0.348; **GxSNPxELS**: F = 0.333; *p* = 0.855 *With ELS (CG:U > B) *.*
**MR-2 G/C CG**	367.28 (1.17)	366.43 (1.50)	445.66 (1.28)	345.14 (1.22)	208.93 (1.27)	184.08 (1.22)	
**MR-2 G/C GG**	412.10 (1.22)	221.31 (1.98)	221.31 (1.33)	202.77 (1.25)	298.54 (1.39)	165.20 (1.35)	
**GR N363S TT**	333.43 (1.13)	307.61 (1.37)	342.77 (1.17)	293.76 (1.17)	255.86 (1.19)	195.43 (1.17)	**G**: F = 0.812; *p* = 0.446; **ELS**: F = 2.205; *p* = 0.140; **SNP**: F = 1.024; *p* = 0.313; **GxELS**: F = 0.178; *p* = 0.837; **GxSNP**: F = 0.174; *p* = 0.840; **SNPxELS**: F = 5.188; *p* = 0.024 * *Without ELS (TC < TT) *;* **GxSNPxELS**: F = 0.489; *p* = 0.485.
**GR N363S TC**	198.15 (1.99)	--	153.10 (1.62)	312.60 (1.49)	86.70 (2.03)	331.13 (1.66)	
**GR N363S CC**	--	--	--	--	--	--	
**GR R22/23 K CC**	341.98 (1.13)	303.39 (1.37)	336.51 (1.16)	293.09 (1.16)	247.17 (1.19)	197.70 (1.16)	**G**: F = 1.069; *p* = 0.346; **ELS**: F = 1.275; *p* = 0.260; **SNP**: F = 2.839; *p* = 0.094; **GxELS**: F = 0.045; *p* = 0.956; **GxSNP**: F = 1.267; *p* = 0.285 *MDD (CT < CC) ** *MDD (without ELS: CT < CC) *.*
**GR R22/23 K CT**	308.32 (1.98)	--	69.66 (1.99)	--	173.38 (1.99)	--	
**GR R22/23 K TT**	--	--	--	--	--	--	
**GR BclI GG**	354.81 (1.17)	516.42 (1.99)	277.33 (1.28)	314.05 (1.24)	277.33 (1.24)	165.20 (1.24)	**G**: F = 2.042; *p* = 0.134; **ELS**: F = 0.394; *p* = 0.531; **SNP**: F = 1.274; *p* = 0.283; **GxELS**: F = 0.323; *p* = 0.725; **GxSNP**: F = 0.282; *p* = 0.889; **SNPXELS**: F = 0.650; *p* = 0.523; **GxSNPXELS**: F = 1.237; *p* = 0.298.
**GR BclI GC**	324.34 (1.17)	285.76 (1.43)	360.58 (1.22)	255.86 (1.23)	170.22 (1.34)	205.59 (1.27)	
**GR BclI CC**	375.84 (1.34)	--	225.94 (2.02)	511.68 (1.68)	272.90 (2.02)	346.74 (1.49)	

Means and standard error (s.e.) were adjusted for covariables (age, depressive symptom intensity score, oral contraceptive, lithium, anticonvulsant, antipsychotic). HC: healthy controls. ELS: early-life stress as independent factor; G: groups as independent factor; GR: glucocorticoid receptor gene; MR: mineralocorticoid receptor gene; SNP: polymorphism independent factor; GxELS: interaction between groups and early-life stress; GxSNP: interaction between groups and polymorphisms; GxSNPXELS: interaction among groups, polymorphisms, and early-life-stress. Post hoc test (SIDAK): * *p* < 0.05.

**Table 4 brainsci-12-01476-t004:** Severity of depressive symptoms in MDD and BD patients concerning history of early-life stress and allelic frequencies of mineralocorticoid and glucocorticoid receptor polymorphisms.

	Unipolar		Bipolar		
	(Mean ± Standard Error)		(Mean ± Standard Error)		
	Without ELS	With ELS	Without ELS	With ELS	
**MR MI180V AA**	7.97 (0.15)	18.27 (0.13)	11.53 (0.11)	17.97 (0.12)	**G**: F = 0.273; *p* = 0.602; **SNP**: F = 2.673; *p* = 0.073 **ELS**: F = 6.855; *p* = 0.010 *; **GxSNP**: F = 0.212; *p* = 0.646; **GxELS**: F = 0.105; *p* = 0.747; **SNPxELS**: F = 1.307; *p* = 0.255; **GxSNPxELS**: F = 2.095; *p* = 0.150 *(MDD without ELS: AA<AG) **
**MR MI180V AG**	19.18 (0.29)	18.27 (0.12)	14.52 (0.38)	24.82 (0.34)	
**MR MI180V GG**	--	8.04 (1.05)	--	--	
**MR-2 G/C** **CC**	6.98 (0.26)	9.59 (0.31)	10.01 (0.31)	14.15 (0.25)	**G**: F = 1.490; *p* = 0.224; **SNP**: F = 4.769; *p* = 0.010 * *(CC < CG*; CC < GG*)*; **ELS**: F = 12.425; *p* = 0.001 *; **GxSNP**: F = 1.489; *p* = 0.229 *MDD (CC < CG) **; **GxELS**: F = 0.004; *p* = 0.948; **SNPxELS**: F = 0.253; *p* = 0.777; **GxSNPxELS**: F = 2.118; *p* = 0.124 *MDD(CG: without ELS < with ELS)** *MDD (without ELS: CC < GG) ** *BD (CG without ELS < with ELS) *.*
**MR-2 G/C** **CG**	8.25 (0.21)	19.70 (0.16)	15.14 (0.22)	19.61 (0.16)	
**MR-2 G/C** **GG**	16.46 (0.10)	19.80 (0.22)	10.53 (0.24)	23.04 (0.24)	
**GR N363S** **TT**	10.17 (0.14)	17.75 (0.12)	11.79 (0.15)	18.45 (0.12)	**G**: F = 1.627; *p* = 0.204; **SNP**: F = 0.034; *p* = 0.854; **ELS**: F = 8.031; *p* = 0.005 *; **GxSNP**: F = 0.897; *p* = 0.345; **GxELS**: F = 0.397; *p* = 0.530; **SNPxELS**: F = 0.834; *p* = 0.363; **GxSNPxELS**: F = 0.186; *p* = 0.667;
**GR N363S** **TC**	5.67 (0.68)	20.58 (0.53)	14.03 (0.53)	27.91 (0.68)	
**GR N363S** **CC**	--	--	--	--	
**GR R22/23 K CC**	9.59 (0.14)	17.92 (0.12)	11.94 (0.14)	13.89 (1.08)	**G**: F = 0.071; *p* = 0.790; **SNP**: F = 0.755; *p* = 0.386; **ELS**: F = 17.660; *p* < 0.001 **;* **GxSNP**: F = 0.357; *p* = 0.551 **GxELS**: F = 0.413; *p* = 0.522;
**GR R22/23 K CT**	21.86 (1.07)	--	18.81 (0.12)	--	
**GR R22/23 K TT**	-	-	-	-	
**GR BclI** **GG**	7.79 (0.22)	15.83 (0.21)	11.22 (0.20)	17.32 (0.18)	**G**: F = 0.016; *p* = 0.900; **SNP**: F = 1.462; *p* = 0.235; **ELS**: F = 10.651; *p* = 0.001 *; **GxSNP**: F = 0.936; *p* = 0.395; **GxELS**: F = 0.020; *p* = 0.889; **SNPxELS**: F = 0.121; *p* = 0.886; **GxSNPxELS**: F = 0.130; *p* = 0.878.
**GR BclI** **GC**	11.19 (0.20)	18.54 (0.16)	13.55 (0.23)	20.98 (0.21)	
**GR BclI** **CC**	14.59 (0.53)	26.35 (0.53)	8.42 (0.71)	18.50 (0.32)	

Means and standard error were adjusted for covariables (anticonvulsant and lithium). ELS: early-life stress; G: group as independent factor; GR: glucocorticoid receptor gene; GRID-HAM-D21: GRID–Hamilton Depression Scale; GxELS: interaction between factors group and early-life stress; SNPxELS: interaction between factors polymorphism and early-life stress; GxSNPxELS: interaction among factors group, polymorphism and early-life stress; MR: mineralocorticoid receptor gene; SNP: polymorphism as independent factor; post hoc test (SIDAK test): * *p* < 0.05.

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
