# Peer review of "Differential Diagnosis of Major Depressive Disorder and Bipolar Disorder: Genetic and Hormonal Assessment and the Influence of Early-Life Stress"

_brainsci, 2022, doi:10.3390/brainsci12111476_

Round 1

Reviewer 1 Report

The paper is an interesting study about the possible role of genetics and hormonal assessments in the differential diagnosis between MDD and BD, taking into care the early-life stress. The manuscript is well written and the methods are clear. However, I think the authors should address some points before considering the manuscript for publication.

- the link line 84 is not working. Moreover, I am not sure about the power analysis because you are including 9 different populations. Please, clarify this aspect. Moreover, the reasons for the inclusion of 9 populations are not clear to me.

- Did MDD and BD patients have stable drug therapy? 

- Which cut-off for the CTQ questionnaire have you applied?

- Have you evaluated if the 3 different subgroups for each diagnosis are statistically similar? 

- please be consistent in your text with the definitions. I know that unipolar and MDD refer to the same population, but different readers might not.

- Please change the references to Supplementary Materials to Appendixes in the text. 

- post hoc analyses are really complicated to be evaluated because it is not clear the polymorphisms' lines

Author Response

Dear Editors Brain Sciences (section "Psychiatric Diseases)".

Dear Dr. Rebecca Strawbridge Special Issue Editor

"Bipolar Disorders: Progressing from Bench to Bedside"

We are delighted that you invited us to resubmit the manuscript: “Differential Diagnosis of Major Depressive Disorder and Bipolar Disorder: Genetic and Hormonal Assessment and the Influence of Early-life Stress” by Menezes IC, Baes CW, Fígaro-Drumond FV, Macedo BBD, Bueno AC, Lacchini R, Mello MF, Castro M and Juruena MF to be published in the Brain Sciences (section "Psychiatric Diseases)" in this Special Issue "Bipolar Disorders: Progressing from Bench to Bedside".

We would like to thank Reviewer 1 for his/her kind words:

“The paper is an interesting study about the possible role of genetics and hormonal assessments in the differential diagnosis between MDD and BD, taking into care the early-life stress. The manuscript is well written, and the methods are clear”.

Thank you very much for the feedback given to our study. This has helped us to reconsider, critically appraise and improve our manuscript.

However, I think the authors should address some points before considering the manuscript for publication.

- the link line 84 is not working.

Thanks for this, we have updated the link that is now working https://dceg.cancer.gov/tools/design/pga

-Moreover, I am not sure about the power analysis because you are including 9 different populations. Please, clarify this aspect. Moreover, the reasons for the inclusion of 9 populations are not clear to me.

Indeed, the present study is two studies in one. Concerning the genetic part: the power analysis calculation is usually done for genetic studies because it considers the frequency (MAF) of the alleles and the prevalence of the disorder in the population. So, it is possible to estimate how many people are going to be needed to reach a specific sample power. We have chosen 0.8 or 80% for our study. Thus, the control sample was meant to be at least 1.5 times higher than the unipolar or bipolar groups. So, for the genetic experiment - 273 subjects (n=113 healthy controls (HC); n=78 MDD patients; and n=82 bipolar patients) - the sample sizes were adequate. 

Concerning the neuroendocrine part: neuroendocrine studies, to present a reliable power of analysis, need smaller samples than genetic studies, and need more caution when selecting the subjects considering the sociodemographic variables (such as medications, age, sex, diagnosis) that should not present difference between control and patient groups - as the cautious sample selection we have done.

Actually, the study was composed of a greater sample (genetic sample), and some of these participants have volunteered to continue participating in the neuroendocrine studies (one part assessed blood aldosterone, and the other part assessed blood cortisol). In the literature, neuroendocrine studies usually present smaller samples than genetic studies.

- Did MDD and BD patients have stable drug therapy? 

Thanks for this important point, we included:

“All patients, MDD and Bipolar are in stable drug therapy in the last 8 weeks.”

- Which cut-off for the CTQ questionnaire have you applied?

Thanks for this we updated including:

“As a criterion for defining the Depressives with Early Stress, we considered the classification of at least one subtype of abuse or neglect as Moderate to Severe or Severe to Extreme in the CTQ scores according to definition Bernstein (18). Participants who obtained scores equivalent to the classification No to Minimal or Low to Moderate in the 5 subtypes of early stress assessed by the CTQ were classified as Depressive Without Early Stress in this study”.

- Have you evaluated if the 3 different subgroups for each diagnosis are statistically similar? 

Yes, we have evaluated the 3 different subgroups for each diagnosis, and they are statistically similar regarding demographic data (age, sex, BMI) and genetics (MR MI180V AA/AG/GG; MR -2G/C CC/CG/GG; GR N363S TT/TC/CC; GR R22/23KCC/CT/TT; GR BclI GG/GC/CC), see details in table 1

- please be consistent in your text with the definitions. I know that unipolar and MDD refer to the same population, but different readers might not.

Thanks for it we reviewed and updated this point

- Please change the references to Supplementary Materials to Appendixes in the text. 

We are now confident to have addressed your concerns. Further details can be found in our manuscript and in the responses to the other reviewer.

We appreciate the time the reviewer has taken to analyze our manuscript. We believe that we have taken on board all the suggestions, which have substantially improved the manuscript. In addition, we have extensively reviewed the manuscript, as you can see in the last version.

We are looking forward to hearing back from you.

Yours sincerely,

Dr Mario F. Juruena, MD MSc Dip CBT MPhil PhD SARCPsych, FBPsychA

Reader in Translational Psychiatry

Department of Psychological Medicine

Centre for Affective Disorders

Institute of Psychiatry, Psychology and Neuroscience

King’s College London

Consultant Psychiatrist South London and Maudsley NHS Foundation Trust

Maudsley Advanced Treatment and ECT Service Consultant Lead- MATS

Reviewer 2 Report

This is a very interesting study investigating the potential role of aldosterone, cortisol, MR and GR gene polymorphisms and early-life as biomarkers of major depressive disorder and bipolar disorder. 

The paper is well-written and of interest for the journal; however, several minor changes are recommended.

Abstract.

1- The study design should be described in the abstract section before describing which hormone factors are potentially biomarkers of MDD, BD.

Introduction.

1-At the introduction section, before describing the differences between major depressive disorder and bipolar disorder, the authors should describe the prevalence of each of them, and the frequency of major depressive episodes in patients with BD.

2-Line 49. Several biological and environmental factors contribute to the etiology of MDD and BD. Which kind of factors? I recommend to introduce something about genetics and environment, and afterwards, focus the introduction on HPA and RAAS axis. 

Methods.

1- The first paragraph should be moved to the 2.1. Participants section (at the end probably).

2-The study design should be described at the beginning of the Methods section, before describing the participants and genetic assessment. 

Results.

1-At the beginning of the results section, I recommend to describe the Table 1. Characteristics are described by "Genetics sample", "Cortisol sample" and "aldosterone sample". It should be described previously to make it more clear.

2-What about the role of gender in the results? Is sex/gender potentially influencing the presented results?

Discussion.

1- Line 329. The authors are describing the limitations of the study. I recommend to build a 4.1.Limitations and Strengths section to include these commentaries.

Conclusions

1-The authors are concluding that aldosterone serum levels may be considered as a potential biomarker of diagnosis. What about the investigation of aldosterone as potential biomarker of treatment response?

Can the authors recommend new avenues of investigation?

Author Response

Dear Editors Brain Sciences (section "Psychiatric Diseases)".

Dear Dr. Rebecca Strawbridge Special Issue Editor

"Bipolar Disorders: Progressing from Bench to Bedside"

We are delighted that you invited us to resubmit the manuscript: “Differential Diagnosis of Major Depressive Disorder and Bipolar Disorder: Genetic and Hormonal Assessment and the Influence of Early-life Stress” by Menezes IC, Baes CW, Fígaro-Drumond FV, Macedo BBD, Bueno AC, Lacchini R, Mello MF, Castro M and Juruena MF to be published in the Brain Sciences (section "Psychiatric Diseases)" in this Special Issue "Bipolar Disorders: Progressing from Bench to Bedside".

We would like to thank Reviewer 2 for his/her kind words:

This is a very interesting study investigating the potential role of aldosterone, cortisol, MR and GR gene polymorphisms and early-life as biomarkers of major depressive disorder and bipolar disorder. 

The paper is well-written and of interest for the journal; however, several minor changes are recommended.

Abstract.

  • The study design should be described in the abstract section before describing which hormone factors are potentially biomarkers of MDD, BD.

Thanks for this we updated it, describing the study design

Introduction.

1-At the introduction section, before describing the differences between major depressive disorder and bipolar disorder, the authors should describe the prevalence of each of them, and the frequency of major depressive episodes in patients with BD.

Thanks for this we updated it, describing the prevalence of each of them, and the frequency of major depressive episodes in patients with BD.

“…MDD in the total NCS-R sample are 16.2% lifetime and 6.6% in the 12 months before the interview.An estimated 4.4% of U.S. adults experience bipolar disorder at some time in their lives..”

2-Line 49. Several biological and environmental factors contribute to the etiology of MDD and BD. Which kind of factors? I recommend to introduce something about genetics and environment, and afterwards, focus the introduction on HPA and RAAS axis. 

Thanks for this we updated it, describing the biological and environmental factors, genetics, HPA axis and RAAS

Methods.

  • The first paragraph should be moved to the 2.1. Participants section (at the end probably).

Thanks for this we updated it, moved to the 2.1. Participants section, at the end.

2-The study design should be described at the beginning of the Methods section, before describing the participants and genetic assessment. 

Thanks for this we updated it, moved to the beginning of the Methods section

Results.

1-At the beginning of the results section, I recommend to describe the Table 1. Characteristics are described by "Genetics sample", "Cortisol sample" and "aldosterone sample". It should be described previously to make it more clear.

Thanks for this we updated it, describing the Table 1. Characteristics are described by "Genetics sample", "Cortisol sample" and "aldosterone sample".

2-What about the role of gender in the results? Is sex/gender potentially influencing the presented results?

For the genetic part that assessed single nucleotide polymorphisms, sex does not influence the results. For the neuroendocrine part that assessed hormonal levels, sex may influence. However, as shown in Table 1, there was no statistical difference among samples considering sex. 

Discussion.

  • Line 329. The authors are describing the limitations of the study. I recommend to build a 4.1. Limitations and Strengths section to include these commentaries.

Thanks for this we updated it, including a 4.1 Limitations and Strengths

Conclusions

1-The authors are concluding that aldosterone serum levels may be considered as a potential biomarker of diagnosis. What about the investigation of aldosterone as potential biomarker of treatment response?

Indeed, this is a new avenue of investigation, assess the aldosterone as potential biomarker of treatment response in a clinical trial.

We included:

Moreover, a new avenue of investigation could be assessing the aldosterone as potential biomarker of treatment response in a clinical trial.

We are now confident to have addressed your concerns and present robust results.

Further details can be found in our manuscript and in the responses to the comments in the response letter to editor and reviewers.

We appreciate the time the reviewer has taken to analyze our manuscript. We believe that we have taken on board all the suggestions, which have substantially improved the manuscript. In addition, we have extensively reviewed the manuscript, as you can see in the last version.

We are looking forward to hearing back from you.

Yours sincerely,

Dr Mario F. Juruena, MD MSc Dip CBT MPhil PhD SARCPsych, FBPsychA

Clinical Senior Lecturer in Translational Psychiatry

Department of Psychological Medicine

Centre for Affective Disorders

Institute of Psychiatry, Psychology and Neuroscience

King’s College London

Consultant Psychiatrist South London and Maudsley NHS Foundation Trust

Maudsley Advanced Treatment and ECT Service Consultant Lead- MATS

Round 2

Reviewer 1 Report

I think the authors have addressed all my concerns.